# Knockdown of Atg7 Induces Nuclear-LC3 Dependent Apoptosis and Augments Chemotherapy in Colorectal Cancer Cells

**DOI:** 10.3390/ijms21031099

**Published:** 2020-02-07

**Authors:** Anna-Lena Scherr, Adam Jassowicz, Anna Pató, Christin Elssner, Lars Ismail, Nathalie Schmitt, Paula Hoffmeister, Lasse Neukirch, Georg Gdynia, Benjamin Goeppert, Henning Schulze-Bergkamen, Dirk Jäger, Bruno Christian Köhler

**Affiliations:** 1National Center for Tumor Diseases, University Hospital Heidelberg, Heidelberg 69120, Germany; anna-lena.scherr@nct-heidelberg.de (A.-L.S.); adam@jassowicz.com (A.J.); patoannaterezia@gmail.com (A.P.); Christin.Elssner@gmx.de (C.E.); lars.ismail1986@googlemail.com (L.I.); nathalie.schmitt@nct-heidelberg.de (N.S.); paula.hoffmeister@googlemail.com (P.H.); dirk.jaeger@nct-heidelberg.de (D.J.); 2Clinical Cooperation Unit Applied Tumor Immunity, National Center for Tumor Diseases and German Cancer Research Center, Heidelberg 69120, Germany; lasse.neukirch@nct-heidelberg.de; 3Institute of Pathology, University Hospital Heidelberg, Heidelberg 69120, Germany; Georg.Gdynia@med.uni-heidelberg.de (G.G.); Benjamin.Goeppert@med.uni-heidelberg.de (B.G.); 4Department of Internal Medicine II, Marien Hospital, Wesel46483, Germany; Henning.Schulze-Bergkamen@prohomine.de

**Keywords:** Atg7, LC3, autophagy, apoptosis, colorectal cancer

## Abstract

Autophagy is a catabolic process that enables cells to degrade obsolete content and refuel energy depots. In colorectal cancer (CRC) autophagy has been shown to promote tumorigenesis through energy delivery in the condition of uncontrolled proliferation. With this study, we aimed at evaluating whether autophagy sustains CRC cell viability and if it impacts therapy resistance. Initially, a colorectal cancer tissue micro array, containing mucosa (*n* = 10), adenoma (*n* = 18) and adenocarcinoma (*n* = 49) spots, was stained for expression of essential autophagy proteins LC3b, Atg7, p62 and Beclin-1. Subsequently, central autophagy proteins were downregulated in CRC cells using siRNA technology. Viability assays, flow cytometry and immunoblotting were performed and three-dimensional cell culture was utilized to study autophagy in a tissue mimicking environment. In our study we found an upregulation of Atg7 in CRC. Furthermore, we identified Atg7 as crucial factor within the autophagy network for CRC cell viability. Its disruption induced cell death via triggering apoptosis and in combination with conventional chemotherapy it exerted synergistic effects in inducing CRC cell death. Cell death was strictly dependent on nuclear LC3b, since simultaneous knockdown of Atg7 and LC3b completely restored viability. This study unravels a novel cell death preventing function of Atg7 in interaction with LC3b, thereby unmasking a promising therapeutic target in CRC.

## 1. Background

Colorectal cancer (CRC) is the third most common malignant neoplasia in humans [1,2]. Despite tremendous progress in therapeutic approaches, patients with distant organ metastases still face a poor prognosis. Conventional poly-chemotherapy remains the backbone of treatment in stage IV patients [3].

Autophagy is an evolutionary highly conserved process by which cells refuel energy [4]. In situations of energy deprivation, autophagy enables cells to digest unnecessary organelles and misfolded proteins in order to maintain their essential metabolism. It has been shown that autophagy contributes to the onset and progression of a variety of diseases, including cancer [5]. Additionally, there is growing evidence for the dependency of cancer cells on autophagy for outgrowth and metastatic spread, indicating a tumor promoting role [6,7]. In the context of CRC, autophagy enhances the resistance towards nutrient deprivation, thereby driving tumorigenicity [8].

There have been numerous autophagy related genes (Atg) described, serving as molecular adaptors in a complex signaling network [9]. The signaling process of autophagy can be divided into several steps ending up in the formation of a double membrane vesicle called the autophagosome. After fusion with a lysosome and subsequent acidification, the engulfed cargo becomes degraded and recycled [10].

Microtubule-associated protein 1 light chain 3 (LC3b) is essential for autophagosome formation and cargo tethering [11]. Atg7 interacts with LC3b and is embedded in early steps of autophagosome formation and it has been identified to play a role in a large variety of physiological and pathological conditions [12]. Recently it has been shown that Atg7 contributes to CRC initiation via shaping a microbiome-dependent immune response [11,13]. The exact mechanism by which Atg7 functions in CRC onset and progression remains elusive.

This study aimed at dissecting the role of autophagy for CRC cells in vitro. We found a central role of Atg7 for CRC cell survival. Furthermore, we could show that the protein function appears to reach beyond its very role in autophagy regulation and highlight the inhibition of Atg7 as a potential addition to standard chemotherapy regimens in CRC treatment [14].

## 2. Results

### 2.1. Atg-7 Expression is Increased in Human CRC 

To evaluate expression levels of relevant autophagy proteins during colorectal carcinogenesis and disease progression, immunohistochemical (IHC) staining of a tissue microarray (TMA) containing mucosa (*n* = 10), adenoma (*n* = 18) and adenocarcinoma (*n* = 49) tissue from patients who underwent surgery was performed. In the TMA, Atg7 expression was found to be significantly upregulated (*p* < 0.01; Figure 1a), whereas Beclin-1 expression was significantly decreased in adenocarcinomas compared to (not matched) normal mucosa (*p* < 0.001, Figure 1a). Expression levels of LC3b and the scaffold protein p62 were unaltered during colorectal carcinogenesis (Appendix A). Figure 1b shows representative pictures of immunohistochemical staining for Atg7 and Beclin-1 on mucosa, adenoma and carcinoma cores of the utilized TMA. In order to evaluate whether the expression levels of key autophagic proteins correlate with the amount of Atg7, tissue spots were assigned to three groups (Atg7 low: ≤4; medium: ≤8; high: >8), based on their IHC score. Neither for LC3b nor for p62 or Beclin-1 a significant dependence on Atg7 expression was found (Appendix A).

### 2.2. Loss of Atg-7 Induces Apoptosis of CRC Cells 

In order to clarify to what extend CRC cells depend on a proper autophagic flux, the key autophagic proteins Beclin-1, Atg7 and Atg12 were targeted by small interfering RNA (siRNA). Downregulation of the respective proteins prevented LC3b conversion and lead to an accumulation of the soluble LC3b-I form. Moreover, knockdown of Atg7 reduced expression levels of Beclin1 and Atg12 (Figure 2a). Interestingly, the overexpression of Atg7 did not lead to an increased autophagic flux (Appendix A). This might be due to the fact that colorectal cancer cells often exhibit high basal autophagy levels per se. For a better quantification of cell death, an additional fluorescence activated cell sorting (FACS) analysis has been performed after 48 h of transfection. Here, 15.3% dead cells were detected in the Atg7 knockdown samples (*p* < 0.001). By contrast, transfection with siRNA against Beclin-1 and Atg12 had no significant effect on CRC cell viability (Figure 2b).

To validate the observed cell death phenotype a second siRNA targeting Atg7 and a second CRC cell line (SW480) has been employed. The efficiency of Atg7 knockdown was found to be comparable with both siRNAs applied (Figure 2c). Flow cytometry as well as inverted microscopy proofed siRNA mediated apoptosis induction with visible morphological hallmarks, such as cell shrinkage and pyknosis, in both cell lines (Figure 2d,e).

### 2.3. Cell Death by Loss of Atg-7 Does not Lead to a Counter-Regulatory Proliferation of CRC Cells or an Increased Immunogenicity

For the question of therapeutic applicability, it is important to know whether targeting Atg7 alters the proliferative capacity of CRC cells or their immunogenicity. To investigate proliferation, a tissue mimicking 3D cell culture system, which is suitable for long term cell culture growth, was applied. Subsequent to a culture time of 96 h, IHC staining for Ki67 was done. This revealed an unalteredly high proliferation index of >90% after knockdown of Atg7 (Figure 3a,b). By immunohistochemical detection of cleaved Caspase 3, which showed significant apoptosis induction in Atg7 siRNA-transfected cells with an average of 15.7% (Figure 3a,b; *p* < 0.001), the phenotype persistence over time was proved (Figure 3a). To validate our finding regarding the persistent proliferation after Atg7 knockdown, we additionally performed a Western blot analysis for the cell cycle regulator Cyclin D1 after siRNA mediated downregulation of Atg7 in HT29 and SW480 cells. The results confirmed our previous finding that silencing of Atg7 in CRC cells did not mitigate their proliferative capacity (Appendix A).

As standard chemotherapeutic agents used in CRC treatment are known to be linked to immunogenic responses [15,16], we analyzed whether the phenotype caused by Atg-7 knockout in CRC cell lines would have an impact on immunogenic cell death (ICD). This was assessed by calreticulin surface expression after siRNA transfection, with oxaliplatin serving as a positive control for the assessment of ICD [17]. HT29 and SW480 cells showed no significant increase in calreticulin expression after siRNA mediated knockdown of Atg-7 when compared to the corresponding control with 1.75% and 3.67% respectively. In contrast, HT29 and SW480 cells treated with oxaliplatin showed a significant increase in calreticulin expression with 21.70% (*p* < 0.05) and 12.60% (*p* < 0.001) respectively (Figure 3c,d).

### 2.4. Nuclear Translocation of LC3b in Atg7 Negative CRC Cells 

Subsequently, we assessed whether a putative cell death preventing function of Atg7 is subject to its downstream targets facilitating protein embedding in autophagy signaling. As a comparison, we treated CRC cells with chloroquine (Cq), an established and clinically applied autophagy-inhibitor, and found no cell death induction in two CRC cell lines (Figure 4b). To confirm a sufficient block of autophagy we stained Cq treated HT29 and SW480 cells for LC3b. During autophagy, a cytosolic form of LC3b (LC3b-I) is conjugated by Atg7 and Atg3 to form LC3b-phosphatidylethanolamine conjugate (LC3b-II), which is recruited to autophagosomal membranes. Cq treatment caused the expected increase of cytosolic LC3b foci indicating late stage autophagy inhibition (Figure 4a; black arrows). Compared to cells under Cq treatment, the subcellular localization of LC3b in HT29 and SW480 cells lacking Atg7 greatly differed. Here, an overall increase in nuclear LC3b was observed (Figure 4a; red arrows). For a better quantification, we performed immunoblotting with separated cytosolic and nuclear fractions and subsequent densitometric analysis. This revealed a 1.6-fold increase of nuclear LC3b in HT29 and a 2.3-fold increase of nuclear LC3b in SW480 cells after Atg7 knockdown (Figure 4c).

### 2.5. LC3b is Indispensable for Apoptosis in CRC Cells Lacking Atg-7

Since we speculated from our results that LC3b may play a role for cell death induction caused by the absence of Atg7, we targeted LC3b utilizing siRNA. By siRNA mediated parallel knockdown of Atg7 and LC3b, we observed a striking result. Additional knockdown of LC3b significantly reduced the number of apoptotic cells, as indicated by lowered levels of cleaved poly ADP ribose polymerase (PARP; Figure 5a). For a better quantification, we measured the DNA fragmentation as a downstream event of apoptotic cell death by FACS analysis. In line with the results obtained by Western blot analysis, the cell death phenotype caused by loss of Atg7 was rescued in the absence of LC3b in HT29 and SW480 cells (Figure 5c).

Figure 5b graphically summarizes the supposed mechanism of LC3b driven cell death. Loss of Atg7 might lead to defective lipidation with phosphytidylethanolamine (PE) and translocation of LC3b into the nucleus, finally initiating classical apoptosis.

### 2.6. Knockdown of Atg7 Leads to Lumor Cell Specific Apoptosis and Augments 5-FU and Irinotecan Induced Cell death in Colorectal Cancer Cells

In scope of a translational setting, it is important to determine tumor cell specificity of the observed cell death induction in the absence of Atg7. Therefore, we utilized non-transformed colon mucosa cells (CCD 841 CoN), in which downregulation of Atg7 was achieved by transfection with a corresponding siRNA (Figure 6a). In contrast to CRC cells, knockdown of Atg7 did not induce cell death in intestinal epithelial cells, as shown by FACS analysis (Figure 6b). This data on tumor cell specificity of Atg7 dependent cell death justified an attempt to combine Atg7 inhibition with cytotoxic therapy.

First, we investigated events of DNA-damage in the absence of Atg7 by measuring phosphorylated Histone H2A (pH2AX) as an indicator for DNA double-strand brakes. Strikingly, phosphorylation of H2AX massively took place in the absence of Atg7 (Figure 6c).

Hence, we evaluated the potential of Atg7 manipulation alongside conventional chemotherapy by treating SW480 and HT29 cells with 5-Fluorouracil (5-FU) and Irinotecan after knockdown of Atg7. In the combinatorial approach, loss of Atg7 significantly enhanced cell death induction of both drugs (Figure 6d). For Irinotecan and 5-FU, the synergistic effect was significant both in HT29 and SW480 cells, with the latter ones expressing the stronger phenotype. For instance, in SW480 the percentage of dead cells was 9% after Atg7 knockdown alone or 12% after treatment with 5 µM Irinotecan alone. By combining both siRNA and Irinotecan, the percentage of dead cells significantly increased to 39.7% (*p* < 0.001, Figure 6d), suggesting a synergistic mode of action.

To counter-check the importance of Atg7 for CRC cell viability in presence of chemotherapeutic agents, Atg7 was overexpressed in HT29 and SW480 cells (Figure 6e). Subsequent to treatment with 5FU or Irinotecan for 48h, cell death induction was determined by FACS analysis and the fold change of cell death increase was calculated to the respective control. This revealed a protective function of Atg7 which significantly attenuated the chemotherapy-induced cell death in both cell lines (Figure 6f).

## 3. Discussion

There is emerging evidence that autophagy promotes carcinogenesis in a context- and tissue-specific manner [18,19]. In CRC cells, autophagy is highly active and promotes resistance towards nutrient deprivation, thereby serving as an energy source for outgrowth and metastatic spread [20]. Furthermore, autophagy plays a pivotal role in providing a resistance mechanism against cytotoxic treatment potentially contributing to treatment failures in standard chemotherapies [21]. In TP53-null CRC cells it has, for instance, been demonstrated that Irinotecan treatment elevates the autophagic rate and diminishes thereby cell death execution [22]. Additionally, a recent report discovered a major role of the central autophagy protein Atg7 in the prevention of dysbiosis-induced colorectal tumorigenesis [13,23].

The aim of this work was to further dissect the function of autophagy relevant proteins in CRC cells and their implications on cytotoxic treatment. First, we demonstrated that the autophagy network is dysregulated in CRC. During the adenoma-carcinoma sequence, we found Atg7 to be upregulated and Beclin-1 to be downregulated, while LC3b and the scaffold protein p62 stayed unaffected. In lung tumors, Atg7 has already been identified as an important factor for sustained tumor cell proliferation at later stages [24]. During the colorectal adenoma-carcinoma sequence, Atg7 is embedded in carcinogenesis in two ways: At early stages, blocking Atg7 mediates a stress response comprising tumor cell growth. In later phases of carcinogenesis, the absence of Atg7 shapes a type 1 interferon-related immune response mediating anti-neoplastic effects [25]. Via regulation of the β3 and γ2 chains of laminin-5, Atg7 seems furthermore involved in CRC metastazation. Interestingly, it has been shown that this function of Atg7 is autophagy-independent, since autophagy inhibitors do not affect laminin-5 expression levels [26]. Several studies showed an association between laminin-5 γ2 chain expression and progression towards a more invasive cancer cell phenotype in colorectal cancer and other tumor entities [27,28]. In a further attempt to dissect Atg7 in the context of colorectal cancer, we demonstrated that a knockdown of Atg7 kills CRC cells via induction of classical apoptotic cell death utilizing the activation of Caspase 3 and the cleavage of PARP, a so far unreported phenotype.

Recently, it has been shown that the loss of Atg7 leads to a stress response only in tumor but not in non-transformed intestinal epithelial cells, potentially opening up a therapeutic window in the context of CRC treatment [13]. Our data proved that Atg7 is dispensable for intestinal epithelial cells, but essential for survival of CRC cells. In other tissues and the hematopoietic system, Atg7 seems indispensable for cellular fate and survival [29,30]. In neurons, diametral findings have been published so far. On the one hand the deletion of Atg7 seems protective against neuronal death in neonatal brain injury [31]. On the other hand, upregulation of Atg7 attenuates degeneration of motor neurons in the context of amyotrophic lateral sclerosis and frontotemporal dementia [32]. Taken together, the role of Atg7 for non-transformed as well as for tumor cells remains controversial and appears to be context and cell-type dependent. Importantly, among the autophagy regulating proteins analyzed, we show that only the absence of Atg7 led to a significant decrease in cell viability. We point out a potentially unique role for Atg7, as neither chemical inhibition with Chloroquine nor loss of Beclin-1 or Atg12 induced cancer cell death, even though autophagy signaling was reliably blocked. The lipidated form of LC3b, termed LC3b-II, stably associates with the autophagosomal membrane. Interestingly, we found that the apoptosis-driving function of Atg7 knockdown depended on the presence of LC3b in the nucleus. A double knockdown of LC3b and Atg7 comprehensively restored the viability of CRC cells. Thereby, LC3b accumulated in the nucleus in the absence of Atg7. Recently, a publication by Kang et al. describes an accumulation of LC3 in the absence of Atg7 and subsequent cell death induction in ovaria [33]. However, in contrast to our work they found LC3 to accumulate at but not in the nucleus. Nevertheless, nuclear translocation of LC3 is a known phenomenon. For instance, it has been described that acetylation of LC3 causes its nuclear translocation, deacetylation upon starvation causes shuttling into the cytosol [34]. Moreover, the deacetylation prior to cytosolic translocation is required for LC3 to bind with Atg7 for its lipidation. In the nucleus, LC3 is bound to several structure proteins [35]. Intriguingly, the specific degradation of nuclear components as an organelle specific recycling may be involved in nuclear-LC3 dependent apoptosis [36,37,38,39]. Our data support the hypothesis that dysregulated autophagy in the absence of Atg7 leads to nuclear LC3 accumulation. In the nucleus, LC3 may exert functions inevitably leading to apoptotic cell death in CRC cells.

With the contribution of autophagy in mediating immunogenic responses from chemotherapeutic agents [40,41], our study further aimed to investigate the immunologic visibility of dying CRC cells after Atg7 knockdown. Here we have shown that the knockdown of Atg7 alone does not lead to an increase in the immunogenicity of apoptotic CRC cells.

Nevertheless, we demonstrated a vulnerability of CRC cells lacking Atg7 and hence, we sought to exploit it in a therapeutic approach. In previous studies, Atg7 has been shown to possess further autophagy-independent functions like modulating the activity of p53 during metabolic stress [42]. It seems to maintain genomic integrity by promoting proper DNA repair via homologous recombination [43]. With regard thereto, we determined the level of pH2AX after knockdown of Atg7 and found it to be significantly increased. This data demonstrated defective DNA-repair in the absence of Atg7, thus requiring further investigation [44]. Furthermore, we have shown that downregulating Atg7 targets CRC cells only, while sparing intestinal epithelial cells. In summary, we considered this as potentially beneficial in the context of cytostatic treatment and applied the chemotherapeutic agents 5-Fluoruracil and Irinotecan, subsequently to Atg7 knockdown, showing a synergistic mode of action. Hence, our data indicate that downregulation of Atg7 sensitizes CRC cells towards Irinotecan and 5-FU treatment, additionally emphasizing the therapeutic potential of targeting Atg7 alone or as a combinatorial approach in CRC.

## 4. Conclusions

Our data indicate that Atg7 plays a unique role within the autophagic machinery of CRC cells: i) Atg7 was found to be upregulated in CRC specimens ii) Atg7 exerts a cell death preventing function in CRC, since its loss induced apoptosis iii) Cell death in the absence of Atg7 strictly depended on nuclear trafficking of LC3 iv) The efficacy of standard chemotherapy against CRC cells was augmented by inhibiting Atg7. Based on our results, further studies targeting Atg7 may open new therapeutic options for CRC.

## 5. Methods

### 5.1. Cell Lines and Chemotherapy

The colorectal cancer cell lines HT29 and SW480 and the non-transformed colon cell line CCD 841 CoN were obtained from ATCC (Manassas, VA, USA) and cultured under standard conditions as described previously, supplemented with 10% fetal bovine serum (PAA laboratories, Cölbe, Germany) and 1% Penicillin/Streptomycin (Sigma-Aldrich, Munich, Germany) in a humidified atmosphere [45]. 5-Fluorouracil (5-FU) and Irinotecan were obtained from Sigma-Aldrich and dissolved in DMSO. Appropriate concentrations and combinatorial treatments are indicated in the figure legends.

### 5.2. RNA-Interference and Plasmid DNA Transfection

Transfections were carried out in OptiMEM (Gibco, Waltham, MA, USA) without supplements using Lipofectamine RNAi-Max (Invitrogen, Karlsruhe, Germany) as described [46]. The following sequences for siRNA targeting were used (MWG Biotech, Ebersberg, Germany): Beclin-1 5′ – AAGAUCCUGGACCGUGUCACC(dTdT) – 3′, ATG-7 #1: 5′ – AUCAGGCACUGCUCUUGAA(dTdT) – 3′, ATG-7 #2: 5′ – GCAC UAGAGUGUGCAUAUG(dTdT) – 3′, LC3b: 5′ – GAUAAUUAGAAGGCGCUUA(dTdT) – 3′, scrambled: 5′ – AGACCCACUCGGAUGUGAAGAGAUA(dTdT) – 3′. Atg12 siRNA (sc-72578) was purchased from Santa Cruz Biotechnology (Heidelberg, Germany). A non-targeting siRNA (siSc) was used as a control. 24 h after transfection, cells were incubated 24 h or 48 h for additional treatment as indicated. Plasmid transfection was performed using Lipofectamine LTX (Invitrogen) in OptiMEM without supplements, according to the manufacturer’s protocol. The pCMV-myc-Atg7 plasmid (#24921) was a gift from Toren Finkel and purchased via Addgene (Watertown, MA, USA) [47]. A concentration of 0.5 µg/mL has been used for transfection with the corresponding empty vector as control.

### 5.3. Flow Cytometry

Flow cytometry analyses were performed using a FACS CANTO II (Becton Dickinson, Franklin Lakes, NJ, USA) as described previously according to the protocol of Nicoletti et al. [45]. Data were analyzed using the BD FACSDIVA^®^ software. Cells in the sub-G1 fraction were depicted as apoptotic.

### 5.4. Calreticulin Exposure Assay

CRC cells were seeded into 24-well plates and transfected or treated as indicated. For the detection of calreticulin exposure, supernatant and cells from each well were collected in FACS tubes. After pelleting (300 × g, 10 min, 4 °C), the supernatant was discarded and cells were stained with an anti-calreticulin antibody (1:100, AbcamCambridge, USA, No. ab2907) for 30 min under the exclusion of light and on ice. Subsequently to a washing step with PBS, CRC cells were stained with an Alexa488-conjugated secondary antibody (Abcam, No. ab150077) for 30 min under the exclusion of light and on ice. After a second washing step with PBS, cells were incubated with DAPI (1 µg/mL) (Thermo Fisher Scientific, Waltham, MA, USA) for 5 min at room temperature and subsequently analyzed by flow cytometry. DAPI-positive cells were considered to be apoptotic and were thus excluded from the gating.

### 5.5. Human Tissues and Ethics Statement

The Tissue Micro Array (TMA), provided by the Tissue Bank of the National Center for Tumor Diseases (NCT, Heidelberg, Germany), consists of 10 spots for normal mucosa, 18 spots for adenoma and 49 spots for adenocarcinoma. Each spot represents a single patient. Paraffin embedded sections were stained with the respective antibody, using NovoLink Polymer detection System (Leica Microsystems, Wetzlar, Germany) according to the manufacturer’s instructions. Negative controls were generated by omitting the primary antibody. The Immunoreactivity Score was calculated by two independent investigators as described before [48], with separate scores for quantity (0–10% = 1, 11–50% = 2, 51–80% = 3, 81–100% = 4) and quality (unstained = 0, weak = 1, moderate = 2, strong = 3). The following antibodies were used for detection: anti-Atg7 (Cell Signaling Technology, Danvers, MA, USA, No. 8558), anti-p62 (Becton Dickinson, Heidelberg, Germany, No. 610832) and anti-Beclin-1 (GeneTEX, Irvine, CA, USA, No. gtx37770) and anti-LC3b (Cell Signaling Technology, No. 3868). For analysis, scoring was done manually by two investigators. The usage of patient tissue for research purposes was approved by the local ethics committee of the University Hospital of Heidelberg (S-206/2005). All analyses were done anonymously and written informed content was obtained from all donors.

### 5.6. 3D Cell Culture

Long term cell culture in 3-dimensional ALVETEX scaffolds (Reinnervate, Sedgefield, UK) fosters cell-cell-interactions in a tissue mimicking environment and has been described previously [46]. After transfection with siRNA, cells were detached with Accutase^®^ (PAA laboratories, Cölbe, Germany) and placed onto the scaffolds. After 96 h scaffolds were collected, snap frozen and subsequently sectioned and stained as described [46]. The following antibodies were used for detection: anti-Ki-67 (Abcam, Cambridge, USA, No. ab16667), anti-cleaved-Caspase-3 (Abcam, No. ab2302). Five sections per scaffold and five scaffolds per group were analyzed (magnification 20×) in three independent experiments.

### 5.7. Protein Isolation, SDS-PAGE, Densitometry and Western Blot Analysis 

Cell lysis, SDS-page and Western blotting were performed according to standard procedures as described previously [45]. The following antibodies were used: anti-Beclin-1 (Santa Cruz Biotechnology, Heidelberg, Germany, No. sc-48381), anti-Atg12 (Cell Signaling Technology, Danvers, MA, USA, No. 2010), anti-Atg7 (Abcam, Cambridge, USA, No. ab53255), anti-LC3b (Cell Signaling Technology, No. 3868), anti-p62 (Becton Dickinson, Heidelberg, Germany, No. 610832), anti-Tubulin (Sigma, St. Louis, MO, USA, No. T8203-25UL), anti-Lamin B1 (Abcam, No. ab16048), anti-cleaved PARP (Cell Signaling Technology, No. 5625), anti-CyclinD1 (Cell Signaling Technology, No. 2926) and anti-pH2AX (Cell Signaling Technology, No. 9718).

In order to quantify protein expression, we used ImageJ^®^ (by Wayne Rasband at NIH, Bethesda, MD, USA) for densitometric analysis as described previously [49]. In brief, band density was measured relative to the untreated control and then adjusted to tubulin as loading control. For analysis of isolated cytosolic and nuclear fractions, tubulin and Lamin B1 were utilized as loading controls and to demonstrate proper separation.

### 5.8. Statistical Analysis

Statistical analyses were performed using SPSS20^®^ (IBM, Armonk, NY, USA) and GraphPad Prism8^®^ (GraphPad Software, San Diego, CA, USA). In order to analyze continuous scaled endpoints, such as percentage of cell death across group means, an ANOVA was conducted, following a post hoc analysis with multiple pairwise comparison using the Dunnet method. In addition, synergism among chemotherapies and inhibitors was evaluated based on the *p*-value of the interaction effect obtained from a two-way ANOVA. A *p*-value of less than 5% was considered to be significant. (*** *p* < 0.001, ** *p* < 0.01, * *p* < 0.05).

## Figures and Tables

**Figure 1 ijms-21-01099-f001:**
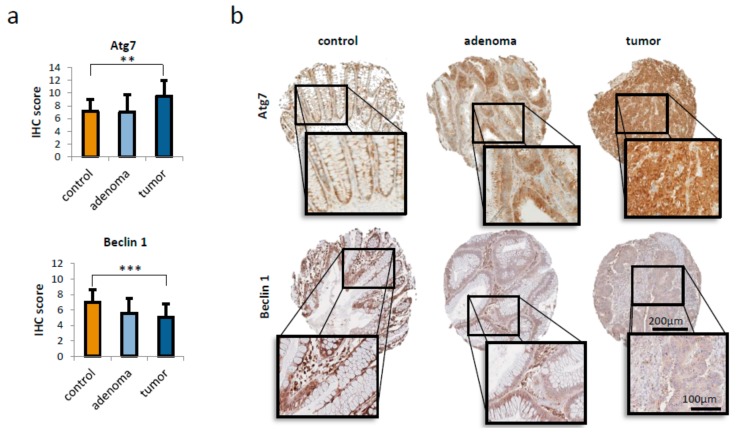
Autophagy regulation in colorectal carcinogenesis. (**a**) Relative expression of autophagy-associated proteins Atg7 and Beclin-1 in a tissue micro array (TMA) of non-matched human colon mucosa (*n* = 10), adenoma (*n* = 18) and carcinoma (*n* = 49). Data represent mean + SD. ** = *p* < 0.01, *** = *p* < 0.001 (**b**) Representative images of Atg7 (upper panel) and Beclin-1 (lower panel) staining on control (mucosa), adenoma and adenocarcinoma TMA cores. Scale bars as indicated.

**Figure 2 ijms-21-01099-f002:**
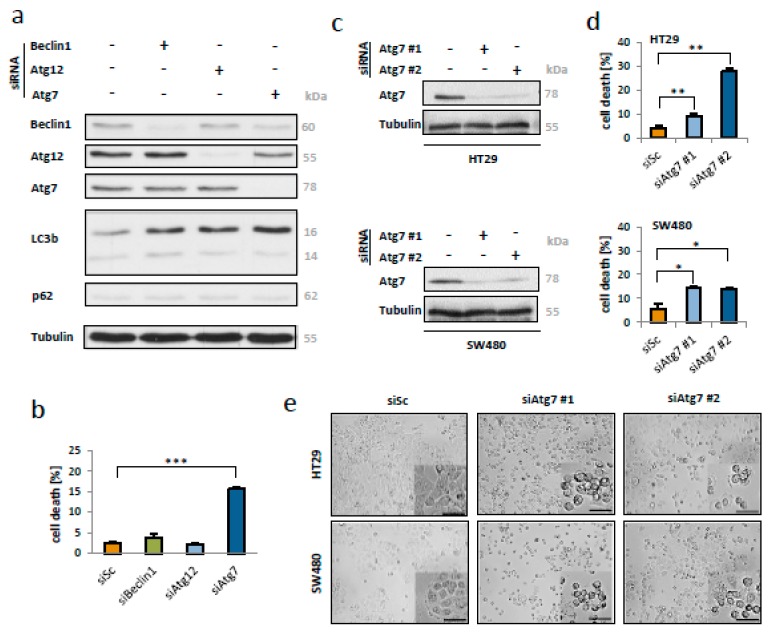
Knockdown of Atg7 but not Beclin-1 or Atg12 induced death of colorectal cancer cells. (**a**) Western blotting for key autophagy proteins after siRNA-mediated knockdown (80 nM) of Beclin-1, Atg12 and Atg7 in HT29 cells. (**b**) Flow cytometry for DNA fragmentation indicating apoptosis after silencing of Beclin-1, Atg12 and Atg7. *** = *p* < 0.001. Data represent mean +SD of independent biological triplicates. (**c**) Western blot analysis for Atg7 after knockdown of Atg7 with two different siRNAs (#1 and #2; 80 nM each) in HT29 and SW480 cells for 48 h. (**d**) Flow cytometry indicating apoptosis induction after transfection with two different siRNAs targeting Atg7 (#1 and #2; 80 nM each) in HT29 and SW480 cells for 48 h. * = *p* < 0.05, ** = *p* < 0.01. (**e**) Bright field microscopy of HT29 and SW480 cells after silencing with two different siRNAs targeting Atg7 (#1 and #2; 80 nM each; scale bar indicates 100 µm).

**Figure 3 ijms-21-01099-f003:**
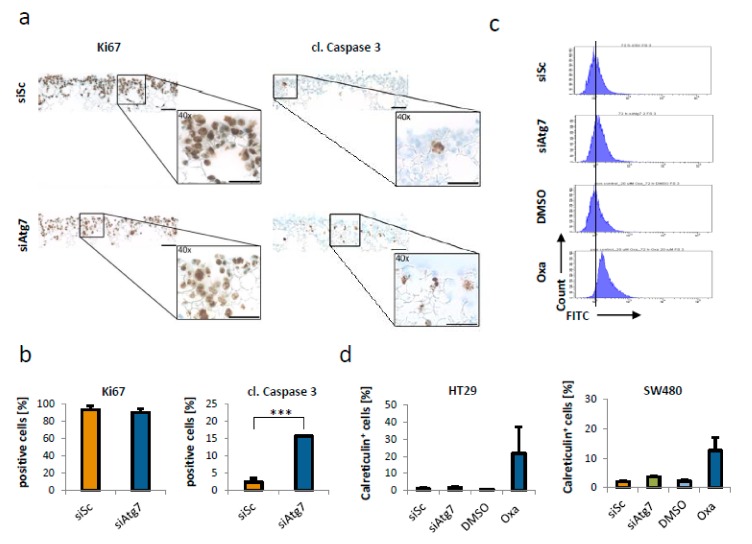
Knockdown of Atg7 did not alter proliferation or immunogenicity of colorectal cancer cells. (**a**) Ki67 (left) and cleaved Caspase 3 (right) staining of siAtg7 (#2; 80 nM) transfected HT29 cells in scaffolds after 96 h. Scale bars indicate 100µm. (**b**) Quantitative analysis corresponding to (**a**). Data represent mean + SD of independent biological triplicates. *** = *p* < 0.001. (**c–d**) Flow cytometry analysis (**c**) and corresponding quantification (**d**) of the surface expression of calreticulin, showing no increase in HT29 and SW480 cells 48 h after transfection with 80 nM of siAtg7. Treatment with 20 µM Oxaliplatin (solved in DMSO) for 48 h served as positive control. Data represent mean + SD of independent biological triplicates. X-axis in (**c**) gives logarithmic fluorescence intensity.

**Figure 4 ijms-21-01099-f004:**
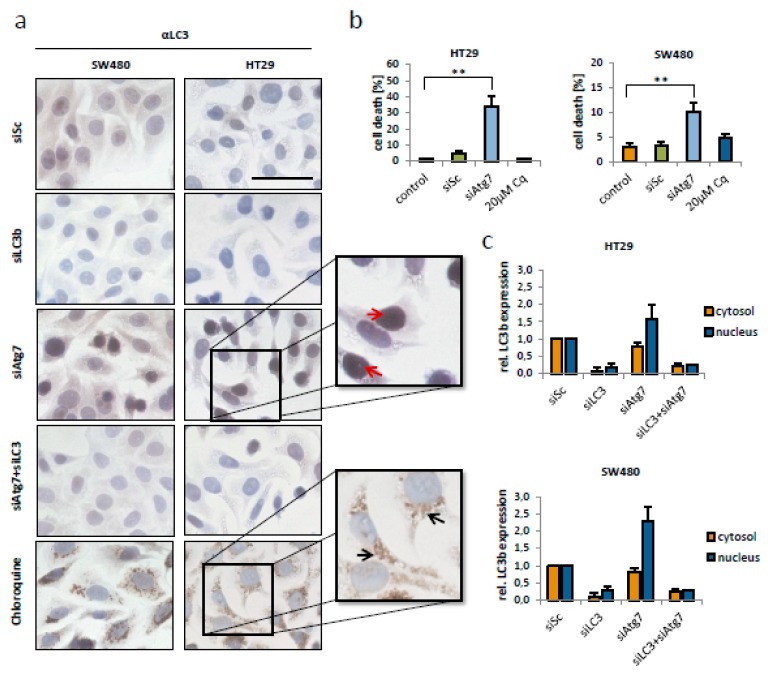
LC3b translocated into the nucleus of colorectal cancer cells after knockdown of Atg7. (**a**) Immunohistochemical staining for LC3b in HT29 and SW480 cells after siRNA-mediated knockdown of Atg7 and/or LC3b (80 nM) and treatment with chloroquine (20 µM) for 48 h. Red arrows indicate nuclear LC3b in siAtg7 transfected cells, black arrows indicate LC3b foci in the cytosol of chloroquine treated CRC cells (scale bar indicates 50 µm). (**b**) Flow cytometry for apoptosis induction after transfection with siRNA targeting Atg7 (80 nM) and treatment with chloroquine (Cq) for 48 h in HT29 (left) and SW480 (right) cells. (**c**) Densitometric analyses of Western blots, quantifying LC3b levels in the cytosol and nucleus of HT29 and SW480 cells, 48 h post transfection with siAtg7 and/or siLC3b (80 nM). Data represent mean + SD of independent biological triplicates.

**Figure 5 ijms-21-01099-f005:**
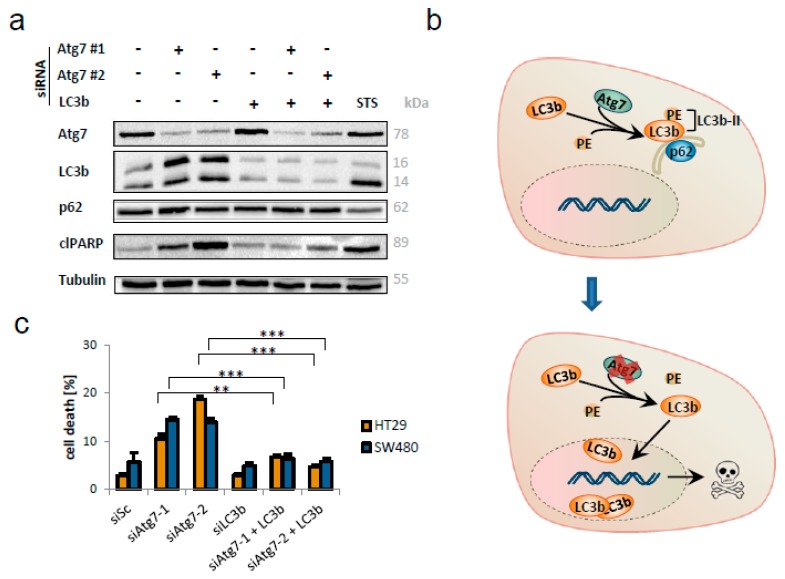
Additional knockdown of LC3b protected colorectal cancer cells from siAtg7-induced apoptosis. (**a**) Immunoblotting of HT29 cells 48 h post-transfection with siRNAs targeting Atg7 (#1 and #2) and/or LC3b (80 nM each). Staurosporine (STS, 2 µM, 4 h) served as positive control for apoptosis induction indicated by cleaved PARP. (**b**) Graphical synopsis of the assumed mechanism of LC3b mediated cell death in CRC cells after knockdown of Atg7. PE = Phosphatidylethanolamine. (**c**) Quantification of cell death in SW480 and HT29 cells, measured by flow cytometry 48 h after transfection with siRNAs targeting Atg7 (#1 and #2) and/or LC3b (80 nM each). Data represent mean + SD of independent biological triplicates. **= *p* < 0.01, ***= *p* < 0.001.

**Figure 6 ijms-21-01099-f006:**
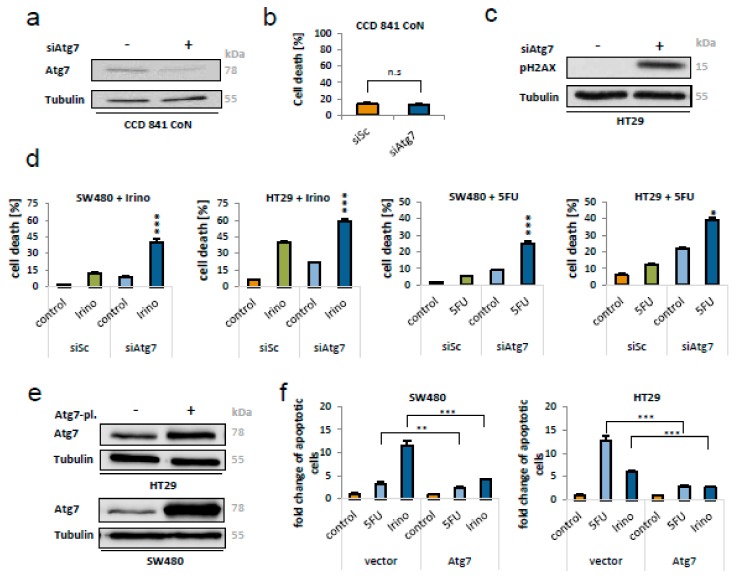
Knockdown of Atg7 led to cancer cell specific apoptosis and augmented chemotherapy in colorectal cancer cells. (**a**) Immunobloting of normal intestinal epithelial cells (CCD 841 CoN) 48 h after siRNA (80 nM) mediated knockdown of Atg7. (**b**) Corresponding FACS analysis, indicating no apoptosis in CCD 841 CoN cells after disruption of Atg7 via siRNA. (**c**) Immunoblotting of pH2AX, indicating DNA damage in HT29 cells after knockdown of Atg7 (80 nM; 24 h). (**d**) Flow cytometry for DNA fragmentation indicating apoptosis after transfection with 80 nM of siAtg7 (#1 for SW480 and #2 for HT29), treatment with 5-FU (5 µg/mL for SW480 and 2 µg/mL for HT29) or the combination of both for 48 h; or the same setting with Irinotecan (5 µM for SW480 and 10 µM for HT29) for 48 h. *p*-values for the interaction indicating a synergistic effect are based on two-way ANOVA; * = *p* < 0.05, *** = *p* < 0.001. Data represent mean + SD of independent biological triplicates. (**e**) Immunoblotting of HT29 and SW480 cells 48 h post-transfection with a plasmid expressing Atg7. (**f**) Quantification of cell death in SW480 (left) and HT29 cells (right), measured by flow cytometry 48 h after transfection with a plasmid expressing Atg7, treatment with either 5-FU (50 µg/mL) or Irinotecan (25 µM) or a combination thereof. ** = *p* < 0.01, *** = *p* < 0.001.

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
