# Peer review of "Knockdown of Atg7 Induces Nuclear-LC3 Dependent Apoptosis and Augments Chemotherapy in Colorectal Cancer Cells"

_ijms, 2020, doi:10.3390/ijms21031099_

Round 1
Reviewer 1 Report
The authors so far addressed all comments and reformatted the text.
I do find that they performed an exhaustive series of experiments, regarding expression up- or down-regulation, that may seem confusing, or even encouraging to request for more. The intention of the paper was not the dissection of the pathway, but the investigation of Atg7 signalling and the integration of the findings into the current schemes, given the two points of the title:
1) Knockdown of Atg7 induced nuclear-LC3 dependent apoptosis and
2) augmentation of chemotherapy
There are still some missing points in the discussion regarding colorectal (cancer) cells.
- The comparative discussion of Atg7 and its role in tissue viability could be related to the lineage type. Colorectal cancer cells express a variety of laminins, and despite the recent findings in the field, this information is missing.
- Irinotecan is converted to SN38 and then inactivated by colorectal cells, thus impeding the full potential of the compound/chemotherapy. I believe that it is essential to discuss the fate of the compound during the experimental procedure: one that could involve a working apoptotic machinery (beclin, nuclear LC3 & topoisomerase), as long as SN38 is present, and second the diminished (?) conversion of the compound.
I should probably be clearer in the beginning. I think that the major attributes of the cells and the compounds in question should be discussed, especially in view of the possible integration of the findings into clinical practice.
Author Response
The authors so far addressed all comments and reformatted the text.
I do find that they performed an exhaustive series of experiments, regarding expression up- or down-regulation, that may seem confusing, or even encouraging to request for more. The intention of the paper was not the dissection of the pathway, but the investigation of Atg7 signalling and the integration of the findings into the current schemes, given the two points of the title:
1) Knockdown of Atg7 induced nuclear-LC3 dependent apoptosis and
2) augmentation of chemotherapy
There are still some missing points in the discussion regarding colorectal (cancer) cells.
- The comparative discussion of Atg7 and its role in tissue viability could be related to the lineage type. Colorectal cancer cells express a variety of laminins, and despite the recent findings in the field, this information is missing.
We thank Reviewer 1 that he turned our attention to this very interesting and apparently also autophagy-independent function of Atg7. A respective section, illuminating the connection between Atg7 and laminin-5 in CRC, has been included in the discussion section and is highlighted in green.
- Irinotecan is converted to SN38 and then inactivated by colorectal cells, thus impeding the full potential of the compound/chemotherapy. I believe that it is essential to discuss the fate of the compound during the experimental procedure: one that could involve a working apoptotic machinery (beclin, nuclear LC3 & topoisomerase), as long as SN38 is present, and second the diminished (?) conversion of the compound.
It has been shown in various publications that HT29 cells have an extremely limited capacity to convert Irinotecan into SN38 in vitro, even though carboxylesterase 2 (CES2) is expressed (Pavillard et al., Cancer Chemotherapy and Pharmacology, 2002 and Wu et al., Clinical Cancer Research, 2002). For SW480 this has not been investigated so far, but a publication evaluating the efficiency of CES2 to hydrolyze other substrates than Irinotecan, points towards the same direction (Wong et al., The Journal of Pharmacology and Experimental Therapeutics, 2012). Even if the efficiency of Irinotecan is significantly lower than that of SN38, it also displays the same inhibitory function. Furthermore, we not solely used the topoisomerase I-Inhibitor Irinotecan but also the pyrimidin analogon 5-Fluorouracil to deduce our conclusion about augmented chemotherapy after Atg7 silencing. Therefore, we believe that the discussion of the fate and turnover of SN38 goes beyond the focus of this manuscript. But we totally agree with Reviewer 1 that our in vitro system does not resemble the in vivo situation in this point and that follow up studies in vivo will be necessary to further dissect the mechanism that leads to an enhanced susceptibility towards chemotherapeutic treatment after loss of Atg7.
I should probably be clearer in the beginning. I think that the major attributes of the cells and the compounds in question should be discussed, especially in view of the possible integration of the findings into clinical practice.
Reviewer 2 Report
This manuscript reports novel results on the function of Atg7 in colon cancer. The experiments were carefully designed and the data are convincing. The observation that knockdown of Atg7 induces LC3 dependent apoptosis is quite important in that Atg7 may represent a good target in colon cancer, especially for sensitizing chemotherapy. Only minor revision is needed before the manuscript can be accepted for publication.
Fig 2, western blot for SW480 should be shown for siRNA knockdown results. Figs 2, 4, 6, the time when cell death was determined should be provided. Fig 6, the sequence and time of siRNA/chemo combination should be explained in detail.Author Response
This manuscript reports novel results on the function of Atg7 in colon cancer. The experiments were carefully designed and the data are convincing. The observation that knockdown of Atg7 induces LC3 dependent apoptosis is quite important in that Atg7 may represent a good target in colon cancer, especially for sensitizing chemotherapy. Only minor revision is needed before the manuscript can be accepted for publication.
- Fig 2, western blot for SW480 should be shown for siRNA knockdown results.
We agree with Reviewer 2 that this information is missing. Accordingly, Western blots proving the efficient knockdown of Atg7 in HT29 and SW480 with both siRNAs applied were included into the manuscript as new figure 2c.
- Figs 2, 4, 6, the time when cell death was determined should be provided. Fig 6, the sequence and time of siRNA/chemo combination should be explained in detail.
We are thankful that Reviewer 2 turned our attention thereto. The incubation times were added in the respective figure legends and are highlighted in green. Furthermore, the siRNA has been specified in figure legend 6 and highlighted in green, as well.
Reviewer 3 Report
The aim of this study is to further examine the role autophagy in CRC and to determine a possible mechanistic role in autophagy in CRC . The authors observed a pattern of increase expression of Atg7 with down regulation of Beclin-1 in tumor compare to NAT. Other common autophagy proteins showed flat expression. The base for targeting Atg7 was determined from this staining of TMA, with successful knockdown showing an increase in cell death. The authors then demonstrated that removal of Atg7 lead to increase in nuclear LC3b leading to cell death. The authors then were able to show and synergist effect between knockdown of Atg7 and common CRC chemotherapy. Overall the authors have constructed a good experimental plan supporting a strong claim.
Pros
Authors have constructed a strong experimental plan in which they are able to connect one figure to another. Each figure is well planned and clearly corroborates the hypothesis. The manuscript is well written with no obvious grammatical errors.
Cons
While the authors have a strong start with determining the role of Atg7 role in autophagy with a knockdown experiment. However, there is no gain of function experiment supporting the hypothesis. Does overexpression of Atg7 leads to greater autophagy and similar downstream effects.
In figure 3 the authors show that proliferation is unaffected when Atg7 is silenced and that it does not affect immunogenicity, but this reviewer finds this claim weak the authors only show a death marker in cleaved- Caspase 3 and proliferation marker Ki67. This reviewer does not show any more immunogenetic markers.
Minor notes
Question for the authors on the use of only SW480 and HT29 cells, is this event observed in other CRC cell lines.
Author Response
The aim of this study is to further examine the role autophagy in CRC and to determine a possible mechanistic role in autophagy in CRC. The authors observed a pattern of increase expression of Atg7 with down regulation of Beclin-1 in tumor compare to NAT. Other common autophagy proteins showed flat expression. The base for targeting Atg7 was determined from this staining of TMA, with successful knockdown showing an increase in cell death. The authors then demonstrated that removal of Atg7 lead to increase in nuclear LC3b leading to cell death. The authors then were able to show and synergist effect between knockdown of Atg7 and common CRC chemotherapy. Overall the authors have constructed a good experimental plan supporting a strong claim.
Pros
Authors have constructed a strong experimental plan in which they are able to connect one figure to another. Each figure is well planned and clearly corroborates the hypothesis. The manuscript is well written with no obvious grammatical errors.
Cons
- While the authors have a strong start with determining the role of Atg7 role in autophagy with a knockdown experiment. However, there is no gain of function experiment supporting the hypothesis. Does overexpression of Atg7 leads to greater autophagy and similar downstream effects.
Indeed it is an elegant approach to validate results from knockdown experiments in a gain-of-function setup. Hence, we evaluated key autophagy proteins LC3b and p62 after Atg7 overexpression. Thereby we found no increase in the autophagic flux, what strongly resembles the finding of Pattinson et al. in cardiomyocytes (Pattison et al., Circulation research, 2011). For a further validation, we correlated the Atg7 expression level in human CRC specimens with the amount of p62, LC3b and Beclin1. The results fortify our previous finding, that an increase of Atg7 in CRC cells does not enhance expression of other key autophagic proteins. Since transformed cells, including CRC cells, often exhibit high basal autophagy levels (Lauzier et al., Scientific Reports, 2019), a further acceleration of the autophagic flux by an increase of Atg7 expression might be limited. We incorporated the mentioned data into the manuscript as figure S2a and b.
- In figure 3 the authors show that proliferation is unaffected when Atg7 is silenced and that it does not affect immunogenicity, but this reviewer finds this claim weak the authors only show a death marker in cleaved- Caspase 3 and proliferation marker Ki67. This reviewer does not show any more immunogenetic markers.
We thank Reviewer 2 for his comment and accordingly validated our finding on proliferation by immunoblotting for the cell cycle regulator Cyclin D1 in HT29 and SW480 cells after siRNA mediated downregulation of Atg7. In line with our previous observations, we found neither in SW480 nor in HT29 cells an altered expression of Cyclin D1 after loss of Atg7. The respective data have been included into the manuscript as figure S2c.
Regarding immunogenicity, we investigated in figure 3c and d Calreticulin as major factor determining anticancer immune responses. In previous works it has been shown that downregulation of Calreticulin abolishes immunogenicity and suppresses phagocytosis of cancer cells by dendritic cells (Zitvogel et al., Clinical Cancer Research, 2010 and Obeid et al., Nature Medicine 2007). Since we obtained a clear result with only our Oxaliplatin treated control turning Calreticulin positive and because we don´t deduce any follow up hypotheses, we think our statement on the immunogenicity of siAtg7 induced CRC cell death is permissible.
Minor notes
- Question for the authors on the use of only SW480 and HT29 cells, is this event observed in other CRC cell lines.
We initially screened four colorectal cancer cell lines (CaCo2, SW480, Colo205 and HT29) with regard to their transfectability. Subsequently, we chose the two cells lines (HT29 and SW480) with the best transfection efficiency and stability to ensure our quality standards. But we totally agree with Reviewer 3 that it would be interesting to expand the set of CRC cell lines and maybe even tumor entities in future analyses.
This manuscript is a resubmission of an earlier submission. The following is a list of the peer review reports and author responses from that submission.
Round 1
Reviewer 1 Report
The authors performed a detailed experimental analysis regarding the potential role of Atg7 in the viability of colorectal cancer cells, the type of induced death, and its implication in palliative care, possibly in combination with current pharmacological approaches.
The manuscript is well written, although the number of the figures is high. Figures are already large and do not allow reading the text. Authors could combine some panels and rearrange information.
There are some minor formatting issues that authors could easily addressed.
L34 palliative care
L149 auf auf ....etc
Regarding the data, in Figure 2A LC3b is increased in total cell extracts, when Atg7 is knocked down. In L145 and forward the subcellular localization of LC3b is described, and it can be assumed that Atg7 is implicated in the upregulation of LC3b, rather than the differential localization. Could the authors explain the mechanism underlying these findings, within the experimental time frame?
Given the information about irinotecan metabolism and its effect on drug active metabolites, in the specific types of cells used in this study, is there any scenario relevant to autophagy or apoptosis?
In the Discussion, there are some references to the implication of Atg7 in the viability of other tissues, and its role in autophagy vs apoptosis. I would advise the authors to check some of the recent publications in the field, and comment accordingly:
https://www.ncbi.nlm.nih.gov/pubmed/31242080
https://www.ncbi.nlm.nih.gov/pubmed/31251935
https://www.ncbi.nlm.nih.gov/pubmed/31265765
and possibly link with the phenotype of colorectal cancer cells:
https://www.ncbi.nlm.nih.gov/pmc/articles/PMC5836818/
Overall, I think that the authors should elaborate on a few points, in view of the practical integration of their findings to CRC management, as already established within the experimental procedure.
Author Response
Reviewer 1:
Authors could combine some panels and rearrange information.The authors agree with Reviewer 1 that the provided information can be condensed. Graphs have been removed, where possible without substantial loss of information, and figures have been rearranged. Precisely the following changes have been conducted: Figure 1: Transfer of the LC3b and p62 graphs to the corresponding IHC pictures in figure S1. Reduction of total figure number by condensation (deletion of figure 2b and 3a) and rearrangement of figures 2-4. Streamlining of result section with conducted changes highlighted in yellow. There are some minor formatting issues that authors could easily addressed. L34 palliative care L149 auf auf ....etc
We are grateful for this hint. The manuscript was carefully revised and the respective mistakes were corrected.
In Figure 2A LC3b is increased in total cell extracts, when Atg7 is knocked down. In L145 and forward the subcellular localization of LC3b is described, and it can be assumed that Atg7 is implicated in the upregulation of LC3b, rather than the differential localization. Could the authors explain the mechanism underlying these findings, within the experimental time frame?
In order to investigate whether the nuclear LC3b staining detected by IHC is caused by an over-all upregulation or a translocation of LC3b from the cytoplasm to the nucleus, nuclear and cytosolic fractions have been isolated. Since LC3b rather decreases in the cytoplasm of both cell lines after Atg7 silencing, we hypothesize that LC3 accumulates in the nucleus (figure 5c). This important point has been addressed in the discussion as well.
Given the information about irinotecan metabolism and its effect on drug active metabolites, in the specific types of cells used in this study, is there any scenario relevant to autophagy or apoptosis?
Indeed the question whether cytotoxic agents, such as Irinotecan and it´s biologically active metabolite SN-38, shape the autophagic response in CRC cells is of great interest. In a previous study, Stanislav and colleagues could already show that Irinotecan enhances the autophagic flux in TP53-null HCT116 CRC cells what diminished cell death induction (Stanislav et al., Anticancer Agents Med Chem, 2013). This finding supports our data showing a synergistic effect of combined Atg7 knockdown and cytotoxic treatment. The respective publication was included in the discussion section and cited accordingly.
In the Discussion, there are some references to the implication of Atg7 in the viability of other tissues, and its role in autophagy vs apoptosis. I would advise the authors to check some of the recent publications in the field, and comment accordingly.
We thank Reviewer 1 to draw our attention onto the role of Atg7 for the survival of other cells and tissues. Even though the work of Lin He and co-workers focusses on non-transformed brain endothelial cells, it highlights, like our own work, that Atg7 can contribute to autophagy-independent pathways. The publication from Donde et al. substantiates once more the complex and tissue- and context-specific functioning of Atg7. Recently, another publication by Kang and colleagues showed a resembling phenotype than we found in CRC cells, in human granulosa cell layers that control oocyte quality. All mentioned publications have been incorporated into the discussion section and are highlighted in yellow.
Reviewer 2 Report
This research study aims to investigate wether autophagy sustains CRC cell viability and if it impacts therapy resistance. The submitted study is hard to understand for the reader:
1) First of all there are to many employed study methods (e.g. tissue microarray, immunoistochemistry, RNA technology, viability assay, flow cytometry, immunoblotting).
2) With special regard to the plan of the study it is not clear. In particular why have 10 mucosa, 18 adenoma and 49 adenocarcinoma spots been studied, in terms of stained expression of essential autophagy, LC3b, Atg7, pg2 and Beclin-1? Is there a previously identified statistical plain to select the above numbers spots?
3) Figure 1B it’s not intellegible. The magnification is not specified. Is this a x10,x20,x40 magnification? The showed figures are to small. A non specific background staining is visible. Furthemore a stromal staining is visible, is it significantly? All these images should be changed and tissue should be showed at low and high magnification.
4) At line 341 I read that tissue for research purpose was approved by the local ethics committee of the University Hospital of Heidelberg, however no protocol numbers has been reported.
Finally, in my opinion all these criticisms should be addressed and the manuscript should be rewrited in a easier manner to understand.
Author Response
First of all there are to many employed study methods (e.g. tissue microarray, immunoistochemistry, RNA technology, viability assay, flow cytometry, immunoblotting).
In the course of figure condensation we streamlined the methodology and removed data received from the viability assay. Further reduction of the methodological spectrum appears not realizable to us, since the utilized IHC staining is a necessity for the analysis of the TMA and because flow cytometry and immunoblotting provide different readout information about the phenotype caused by the siRNA-mediated approach.
With special regard to the plan of the study it is not clear. In particular why have 10 mucosa, 18 adenoma and 49 adenocarcinoma spots been studied, in terms of stained expression of essential autophagy, LC3b, Atg7, pg2 and Beclin-1? Is there a previously identified statistical plain to select the above numbers spots?
The tissue micro array (TMA) used in this study was designed by trained pathologists of the department of pathology, University Hospital Heidelberg (Georg Gdynia) and was provided by the tissue bank of the National Center of Tumor Diseases. The TMA has been described and utilized for previous publications (Jia et al., Clinical Epigenetics, 2019; Scherr et al., Cell Death & Disease, 2016). It´s tissue compilation is impartial from statistical considerations serving as an approach to compare unmatched mucosa, adenoma and carcinoma tissue.
Figure 1B it’s not intellegible. The magnification is not specified. Is this a x10,x20,x40 magnification? The showed figures are to small. A non specific background staining is visible. Furthemore a stromal staining is visible, is it significantly? All these images should be changed and tissue should be showed at low and high magnification.
In order to ensure a distinguished image quality, the slides were digitized with an Aperio whole slide scanner (Leica Biosystems, Wetzlar, Germany) upon staining. Therefore, scale bars were included to provide a visual indication of size and to further illustrate that high and low magnifications of the same display detail are depicted. By providing scale bars instead of magnifications the authors furthermore oriented at the current guidelines of many journals. The mentioned staining of stromal cells is specific, since autophagy-relevant proteins, including Atg7 and Beclin-1, are expressed in fibroblasts. However, for the analysis depicted in figure 1a, only CRC cells were taken into account.
At line 341 I read that tissue for research purpose was approved by the local ethics committee of the University Hospital of Heidelberg, however no protocol numbers has been reported.
We thank the Reviewer for drawing our attention to the missing information. We amended the Material&Methods section with the number of the respective ethics vote.
Finally, in my opinion all these criticisms should be addressed and the manuscript should be rewrited in a easier manner to understand.
We thank Reviewer 2 for this comment and accordingly condensed the provided information as outlined above. Furthermore, we rewrote the text thoroughly to achieve a better perspicuity therewith.
Reviewer 3 Report
This study described the functional significance of Atg7 in CRC tumorigenesis. Especially, the authors showed that Arg7 functions as a key mediator of inducing apoptosis via regulating autophagy in a cell type specific manner. In addition, Atg7 inhibition in combination with conventional chemotherapy increases the cytotoxicity in CRC cells, providing its clinical relevance.
Overall, this study is well written and quite interesting. I have a few suggestions to improve the quality of the current manuscript.
Most of the results were based on the "loss-of-function" approaches. It would be necessary to perform additional "gain-of-function" study to confirm some of the conclusions. For example, it is curious whether overexpression of Atg7 indeed renders the chemosensitive CRC cells to resistant to it, which enhance the clinical significance of this study. It is necessary to discuss the genomic alterations of Atg7 and LC3b in CORD and if possible, any association of their expression with survival in CRC patients. Based on the results, the stoichiometric ratio of Atg7 and LC3b in CRC cells seems important in autophagy and consequent apoptosis. Any thoughts for this point?Author Response
Most of the results were based on the “loss-of-function” approaches. It would be necessary to perform additional “gain-of-function” study to confirm some of the conclusions. For example, it is curious whether overexpression of Atg7 indeed renders the chemosensitive CRC cells to resistant to it, which enhance the clinical significance of this study.
The authors thank Reviewer 3 for this valuable suggestion. Accordingly, Atg7 was overexpressed in HT29 and SW480 cells before treatment with chemotherapeutic agents 5FU or Irinotecan. Strikingly, and in line with our data, the increase of Atg7 renders cells less susceptible towards cytotoxic substances. The respective results were included into figure 6 e and f.
It is necessary to discuss the genomic alterations of Atg7 and LC3b in CORD and if possible, any association of their expression with survival in CRC patients.
The authors agree that the question whether Atg7 and LC3b are coordinately expressed is of great interest in the context of this work. By using the co-regulation database (CORD), as recommended by Reviewer 3, we neither found a concordant nor discordant correlation of expression. In order to maintain readability and length of the manuscript, we did not incorporate this into the revised manuscript. We furthermore appreciate the impulse to correlate the Atg7 expression level and overall survival of patients to examine whether Atg7 could serve as a prognostic marker in CRC treatment. However, this goes beyond the resources of this revision and will be addressed in a follow up project.
Based on the results, the stoichiometric ratio of Atg7 and LC3b in CRC cells seems important in autophagy and consequent apoptosis. Any thoughts for this point?
We hypothesize from our data that loss of Atg7 leads to the accumulation of LC3 in the nucleus which causes the observed cell death phenotype, as depicted in figure 5b. In this sense, rather the translocation of LC3 than the increased stoichiometric ratio of LC3/Atg7 seems important. We conclude that the negative correlation between Atg7 and LC3 is not proportional and cell death is initiated if a critical threshold in nuclear LC3 is reached. Based on our observation of an unaltered CRC cell viability after siRNA-mediated knockdown of Beclin-1 and Atg12 or chemical autophagy inhibition with Chloroquine, we conclude that the initiated cell death in Atg7 knockdown cells is an autophagy-independent phenomenon. Interestingly, resembling results have been published by Kang and colleagues, recently (Kang et al., Biochem Biophys Rep, 2018). They report that a loss of Atg7 leads to the accumulation of LC3 with subsequent cell death induction in human cumulus cells. The mentioned publication has been incorporated into the discussion section and is highlighted in yellow.